:̇◯: PLOS | ONE

**Data Availability Statement:** All relevant data are within the manuscript, supporting information files, and through the following relevant accession

# Influence of heat stress on reference genes stability in heart and liver of two chickens genotypes

Juliana Gracielle Gonzaga Gromboni◉[1]*, Haniel Cedraz de Oliveira[2], Daniele Botelho Diniz Marques◉[2], Antônio Amândio Pinto Garcia Junior[3], Ronaldo Vasconcelos Farias Filho[3], Caio Fernando Gromboni[4], Teillor Machado Souza[5], Amauri Arias Wenceslau[6]

**1** Pos graduate program in Animal Science, Universidade Estadual de Santa Cruz—UESC, Ilhéus, BA, Brazil, **2** Department of Animal Science, Universidade Federal de Viçosa, Viçosa, MG, Brazil, **3** Department of Rural and Animal Technology, Universidade Estadual do Sudoeste da Bahia, Itapetinga, BA, Brazil, **4** Instituto Federal de Educação, Ciência e Tecnologia da Bahia–IFBA, Ilhéus, BA, Brazil, **5** Bachelor student of Veterinary Medicine, Universidade Estadual de Santa Cruz—UESC, Ilhéus, BA, Brazil, **6** Department of Agricultural and Environmental Sciences, Universidade Estadual de Santa Cruz—UESC, Ilhéus, BA, Brazil

* jugracielle@hotmail.com

## Abstract

### Introduction

Real-time polymerase chain reaction (RT-qPCR) is an important tool for analyzing gene expression. However, before analyzing the expression of target genes, it is crucial to normalize the reference genes, in order to find the most stable gene to be used as an endogenous control. A gene that remains stable in all samples under different treatments is considered a suitable normalizer. In this sense, we aimed to identify stable reference genes for normalization of target genes in the heart and liver tissues from two genetically divergent groups of chickens (Cobb 500® commercial line and Peloco backyard chickens) under comfort and acute heat stress environmental conditions. Eight reference genes (*ACTB*, *HPRT1*, *RPL5*, *EEF1*, *MRPS27*, *MRPS30*, *TFRC* and *LDHA*) were analyzed for expression stability. The samples were obtained from 24 chickens, 12 from the backyard Peloco and 12 from the Cobb 500® line, exposed to two environmental conditions (comfort and heat stress). Comfort temperature was 23°C and heat stress temperature was 39.5°C for one hour. Subsequently, the animals were euthanized, and heart and liver tissue fragments were collected for RNA extraction and amplification. To determine the stability rate of gene expression, three different statistical algorithms were applied: BestKeeper, geNorm and NormFinder, and to obtain an aggregated stability list, the RankAgregg package of R software was used.

### Results

The most stable genes using BestKeeper tool, including the two factors (genetic group and environmental condition), were *LDHA*, *RPL5* and *MRPS27* for heart tissue, and *TFRC*, *RPL5* and *EEF1* for liver tissue. Applying geNorm algorithm, the best reference genes were *RPL5*, *EEF1* and *MRPS30* for heart tissue and *LDHA*, *EEF1* and *RPL5* for liver. Using the NormFinder algorithm, the best normalizer genes were *EEF1*, *RPL5* and *LDHA* in heart, and *EEF1*,

numbers: ACTB: https://www.ensembl.org/Gallus_
gallus/Transcript/Summary?g=ENSGALG0000
0009621;r=14:4573328-4577556;t=ENSGALT0000
0015673. HPRT1: https://www.ncbi.nlm.nih.gov/
nuccore/5420366. MRPS27: https://www.ncbi.
nlm.nih.gov/nuccore/XM_424803. TFRC: http://
www.ensembl.org/Gallus_gallus/Gene/Summary?
g=ENSGALG00000007485;r=9:15091718-
15104420. EEF1: https://www.ncbi.nlm.nih.gov/
nuccore/NM_204157. MRPS30: https://www.ncbi.
nlm.nih.gov/nuccore/NM_204939.1?report=
GenBank. RPL5: https://www.ncbi.nlm.nih.gov/
nuccore/NM_204581.4. LDHA: https://www.
ensembl.org/Gallus_gallus/Gene/Summary?g=
ENSGALG00000006300;r=5:12754029-12780561.

**Funding:** This study was financially supported by
Fundação de Amparo à Pesquisa no Estado da
Bahia (FAPESB), Universidade Estadual de Santa
Cruz (UESC) and Universidade Estadual do
Sudoeste da Bahia (UESB).

**Competing interests:** The authors have declared
that no competing interests exist.

*RPL5* and *ACTB* in liver tissue. In the overall ranking obtained by RankAggreg package, considering the three algorithms, the *RPL5*, *EEF1* and *LDHA* genes were the most stable for heart tissue, whereas *RPL5*, *EEF1* and *ACTB* were the most stable for liver tissue.

## Conclusion

According to the RankAggreg tool classification based on the three different algorithms (BestKeeper, geNorm and NormFinder), the most stable genes were *RPL5*, *EEF1* and *LDHA* for heart tissue and *RPL5*, *EEF1* and *ACTB* for liver tissue of chickens subjected to comfort and acute heat stress environmental conditions. However, the best reference genes may vary depending on the experimental conditions of each study, such as different breeds, environmental stressors, and tissues analyzed. Therefore, the need to perform priori studies to assay the best reference genes at the outset of each study is emphasized.

## Introduction

Real-time quantitative polymerase chain reaction (RT-qPCR) is a widely used technique for gene expression studies, since it is a sensitive and efficient method for transcript analysis [1,2]. As RT-qPCR is a technique that allows small dynamic changes in gene expression between samples, care must be taken at each step of sample preparation. Therefore, it is necessary to correct the technical variations during the extraction and reverse transcription procedures in order to obtain reliable expression results [3,4]. The most appropriate procedure for expression normalization is to select one or more reference genes (RG) with stable expression [5].

A reference gene indicated as a normalizer should have stable expression levels independent of the treatments and experimental conditions, in order to control the variations that may influence the results [6]. Reference genes commonly used as normalizers are part of the cell structure or participate in essential cell pathways, such as *GAPDH* (glyceraldehyde 3-phosphate dehydrogenase), *18S* and *28S* (ribosomal RNAs), *ACTB* (beta actin), *RPL5* (ribosomal protein L5), among others [7]. Therefore, RG expression stabilities need to be evaluated before the use of these genes as target genes normalizers [8, 9, 10], which can be performed by specific statistical tools, e.g. BestKeeper [11], geNorm [12] and NormFinder [13] algorithms. These tools allow analyzing expression data obtained by RT-qPCR, evaluate the stability of tested RG and indicate the most appropriate gene.

Several studies have demonstrated suitable RG as target genes normalizer in different tissues and species, such as cattle [14] pigs [15], fish [16, 17] and chickens [18, 19, 20, 21]. However, to our knowledge, there are no studies evaluating RG in the heart and liver of backyard chickens raised under an extensive production system, and commercial Cobb 500® chicken, under comfort and acute heat stress environmental conditions.

In this sense, we aimed to identify stable reference genes for normalization of target genes in the heart and liver tissues from two genetically divergent groups of chickens (Cobb 500® commercial line and Peloco backyard chickens) under comfort and acute heat stress conditions.

## Material and methods

### Ethical approval

All experimental procedures were approved by the Ethics Committee on Animal Use—CEUA/UESB, protocol number 109/2015.

## Animals

Twenty four males and females birds (*Gallus gallus domesticus*) were used in the experiment: 12 Peloco chicks (6 controls and 6 heat-stressed) and 12 Cobb 500® commercial chicks (6 controls and 6 heat-stressed). Commercial birds were purchased one week after the hatch of Peloco birds in the poultry sector of Universidade Estadual do Sudoeste da Bahia (UESB), Itapetinga Campus, where they were raised under the same environmental conditions from November 2 to December 2, 2015, with average local temperature of 26.5˚C. Birds were fed following their nutritional requirements, according to Rostagno et al. [22]. All birds were raised in open stalls lined with wood shavings.

## Heat stress

Heat stress was applied in two moments in order to equalize the birds euthanization age (30 days). Firstly, six Peloco birds were subjected to heat stress under an average temperature of 39.5˚C and relative humidity of 60% for one hour, with *ad libitum* access to water and feed. Six Cobb 500® birds were subjected to heat stress under the same conditions. During the whole period of heat stress, birds' behavior was constantly observed. In this way, the birds could be removed from the heat stress condition before dying. When the animals presented prostration and accelerated respiratory rate, the heat stress period was ended and the birds were immediately euthanized by cervical dislocation. Control birds (six from each genetic group) were euthanized early in the morning (4 am at local time) to ensure thermal comfort temperature (23˚C).

## Tissue sampling and RNA extraction and quantification

After euthanization, heart and liver tissues were collected and stored in cryogenic tubes, identified and immediately frozen in liquid nitrogen. After collection, the samples were sent to the Veterinary Genetics Laboratory of Universidade Estadual de Santa Cruz (UESC), where they were stored in an ultrafreezer (-80˚C). Initially, total RNA was extracted from the selected samples with Trizol®, following the manufacturer's protocol. The concentration of the extracted RNA was verified by NanoDrop 2000® spectrophotometer and quality was checked by analyzing rRNA bands after the samples stained with ethidium bromide were submitted to 1% agarose gel electrophoresis, confirming their integrity.

## Reverse transcription of mRNA

Reverse transcription was performed with the commercial kit GoScript TM Reverse Transcription System (Promega Corporation, Madison, USA). Two μl of RNA obtained from tissues (liver and heart), 10μl 10X RT buffer, 4μl dNTP, 1μl Oligodt, 1μl Reverse Transcriptase enzyme, 1μl recombinant ribonuclease inhibitor RNase OUT and ultrapure water were used, completing a final volume of 20μl. The samples were incubated in a thermocycler at 50˚C for 50 min, 85˚C for 5 min, and then chilled on ice. The cDNA obtained was stored in a -20˚C freezer for further gene expression analysis.

## Reference genes selection and RT-qPCR optimization

Eight gene sequences were selected based on their biological and metabolic functions [19] for expression stability analysis, needed in further RT-qPCR studies (Table 1). To calculate PCR efficiency, a standard curve was constructed from a cDNA pool of all treatments and factors for both liver and heart tissues. For this, the following dilutions were performed: 5, 15, 45 and 135ng/μl with three primer concentrations: 200, 400 and 800 mM.

**Table 1. Description of *Gallus gallus* reference genes and their specific primers used in RT-qPCR analyses.** All primers were designed by Nascimento et al. [19].

| GENE | GENE ID | SEQUENCE 5' - 3' | DESCRIPTION |
|------|---------|------------------|-------------|
| ACTB | ENSGALT00000015673 | | Beta actin |
| | | F: ACCCCAAAGCCAACAGA | |
| | | R: CCAGAGTCCATCACAATACC | |
| HPRT1 | AJ132697 | F: GCACTATGACTCTACCGACTATT | Hypoxanthine phosphoribosyltransferase 1 |
| | | R: CAGTTCTGGGTTGATGAGGTT | |
| MRPS27 | XM_424803 | F: GCTCCCAGCTCTATGGTTATG | Mitochondrial ribosomal protein S27 |
| | | R: ATCACCTGCAAGGCTCTATTT | |
| TFRC | ENSGALE000000070489 | F: CTCCTTTGAGGCTGGTGAG | Transferrin Receptor |
| | | R: CGTTCCACACTTTATCCAAGAAG | |
| LDHA | ENSGALG00000006300 | F: CTATGTGGCCTGGAAGATCAG | Lactate Dehydrogenase A |
| | | R: GCAGCTCAGAGGATGGATG | |
| EEF1 | NM_204157.2 | F: GCCCGAAGTTCCTGAAATCT | Eukaryotic translation elongation factor 1 alpha 2 |
| | | R: AACGACCCAGAGGAGGATAA | |
| MRPS30 | NM_204939.1 | F: CCTGAATCCCGAGGTTAACTATT | Mitochondrial ribosomal protein S30 |
| | | R: GAGGTGCGGCTTATCATCTATC | |
| RPL5 | NM_204581.4 | F: AATATAACGCCTGATGGGATGG | Ribosomal protein L5 |
| | | R: CTTGACTTCTCTCTTGGGTTTCT | |

For the comparison of the cycle threshold (Ct) parameters, all tissue samples from both genetic groups in the comfort and heat stress environmental conditions were added to all plates and amplified in duplicates, using RT-qPCR technique. Amplification reactions were performed in a thermocycler, in ddCt assay (Relative Quantification). The RT-qPCR reaction conditions were defined with initial denaturation at 95˚C for two minutes and 40 denaturation cycles at 95˚C for 15 seconds. The extension temperature was individually standardized for each primer pair for 60 seconds (Table 1). At the end of the amplification reaction, an additional step with gradual temperature rise from 60˚C to 95˚C was included to obtain the dissociation curve. Amplification of all genes was performed on the Real Time PCR 7500 Fast System (Applied Biosystems, Foster City, CA, USA) and results were obtained with Sequence Detection Systems software (V.2.0.6) (Applied Biosystems, Foster City, CA, USA), which generated the Ct parameter. The Ct values were obtained directly by the above-mentioned program and used to calculate the average Ct and the standard deviation (SD). The PCR amplification efficiency was calculated for each reference gene using the formula: $E = (10^{\,(-1/angular\ coefficient)\ -1})$ x100 [11].

Subsequently, the dissociation curves were evaluated based on amplification and specificity. After efficiency analysis, the most suitable annealing temperature and primer concentration were used in PCR reactions.

## Real-time quantitative PCR

After calculating the efficiency values and choosing the best parameters (annealing temperature and primer and cDNA concentrations), the samples were subjected to RT-qPCR amplification following the same reaction and thermocycling conditions as the efficiency test. All samples were performed in duplicate.

## Determination of reference genes expression stability

To determine RG stabilities, average Ct values were used as input files in three different statistical algorithms: BestKeeper [11] geNorm [12] and NormFinder [13]. All analyses were performed in R software [23] using the endogenes pipeline (https://github.com/hanielcedraz/refGenes).

In the BestKeeper approach, intergene relationship, Pearson correlation coefficient, sample integrity, and expression stability were calculated for each reference gene by intrinsic expression variation. From these data, the variance of Pearson correlation coefficient was calculated and used for paired correlation analysis between genes. In this way, the gene with the least variation was considered the best normalizer [11].

The calculation process in the geNorm algorithm is based on normalized Ct values, in which the individual values of a gene are normalized to the sample with the lowest Ct value for that gene. In this approach, the pair variation of a specific gene is characterized with all other genes with the SD of expression ratios logarithmically transformed. Gene stability (M) is determined as the average of a gene pair variation with other RG. The gene with the lowest M value is the most stable. To choose the best RG, the geNorm method calculates the M stability after deletion of the less stable gene and repeats the analysis until only the two most stable genes remain. The geNorm approach also determines the minimum number of RG required for proper data normalization [12].

NormFinder is an approach that uses normalized Ct values and estimates the total variation and the variation between subgroups of the same samples. This method applies intra and inter-group variations to calculate a stability value for each gene. Then, candidate RG can be classified based on their stability values, in which the lowest values correspond to the most stable genes [13, 24].

For each tool (BestKeeper, geNorm and NormFinder), a stability ranking considering each factor (genetic group—Peloco and Cobb 500®—and environment—comfort and acute heat stress), and an overall stability ranking (considering all factors) were constructed. An aggregate list was obtained in RankAggreg package [25] of R software using the brute force algorithm with BruteAgregg function for the overall stability rankings considering each statistical tool.

## Results

### Primers efficiencies and specificities

The reaction efficiency test was performed to verify the primers main features prior to RT-qPCR analysis. Primers annealing temperatures ranged from 60 to 62˚C, and cDNA and primer concentrations of 45ng/μl and 400mM, respectively, resulted in the best efficiency for both heart and liver tissues. Amplification efficiency ranged from 95% to 102%, indicating suitable linear correlation. Primer specificity was evaluated by the dissociation curve, with no primer dimers detected.

### Descriptive statistics of reference genes

Eight reference genes were analyzed using the RT-qPCR technique. According to BestKeeper descriptive statistics, there was great expression variability among genes in the liver and heart of the two different genetic groups (Tables 2–5). For the Cobb 500® commercial line, in the heart tissue, the *RPL5* and *ACTB* genes were highly expressed, since presented Ct values of 17.12 and 17.36, respectively. The *HPRT1*, *MRPS27*, *LDHA*, *MRPS30*, *EEF1* and *TFRC* genes presented moderate expressions ranging from 24 to 29 cycles [19]. High coefficient of variations (CV) were also noted, wherein *HPRT1* and *MRPS30* genes presented the highest (6.77%) and lowest (2.75%) values, respectively, for heart tissue. Regarding the SD, a suitable normalizer gene should present SD below 1.0 [11]. Thus, three genes were considered stable in Cobb 500® for this tissue: *RPL5*, *MRPS30* and *EEF1* (SD = 0.73; 0.75 and 0.73, respectively). The other genes presented SD above 1.0 (Table 2).

In the heart tissue of Peloco genetic group, *RPL5*, *ACTB* and *EEF1* genes were highly expressed, with Ct values of 17.25; 19.85 and 21.18, respectively.

**Table 2. Descriptive statistics of reference genes expression levels obtained by BestKeeper in the heart tissue of Cobb 500® commercial line submitted to comfort (23˚C) and acute heat stress (39.5˚C) conditions.**

| n = 12 | RPL5 | LDHA | MRPS30 | EEF1 | ACTB | HPRT1 | MRPS27 | TFRC |
|---|---|---|---|---|---|---|---|---|
| Geometric mean [Ct] | 17.12 | 25.88 | 27.34 | 27.34 | 17.36 | 24.69 | 24.71 | 29.62 |
| Arithmetic mean [Ct] | 17.15 | 25.91 | 27.36 | 20.83 | 17.41 | 24.75 | 24.75 | 29.66 |
| Min [Ct] | 15.58 | 23.90 | 25.46 | 19.93 | 15.68 | 22.39 | 22.52 | 27.19 |
| Max [Ct] | 20.03 | 28.98 | 29.34 | 24.27 | 20.68 | 27.07 | 27.38 | 32.57 |
| Standard deviation [± Ct] | 0.73 | 1.04 | 0.75 | 0.73 | 1.14 | 1.67 | 1.16 | 1.16 |
| CV [%Ct] | 4.28 | 4.02 | 2.75 | 3.52 | 6.56 | 6.77 | 4.68 | 3.91 |
| Correlation coeff. [r] | 0.86 | 0.92 | 0.71 | 0.88 | 0.54 | 0.58 | 0.62 | 0.46 |

[Ct]: Cycle threshold; Min [Ct] and Max [Ct]: Cycle threshold minimum and maximum values, respectively; CV [%Ct]: Coefficient of variation of Ct levels in percentage; [r]: correlation coefficient.

The other genes had expressions ranging from 24 to 29 cycles, thus presenting moderate expressions. In the same tissue, low CV were observed in comparison with Cobb 500®. The *TFRC* and *ACTB* genes presented the lowest and highest CV values (3.61% and 5.08%, respectively). Regarding SD measures, the genes *RPL5*, *EEF1* and *HPRT1* were considered stable in the heart tissue of Peloco genetic group (SD = 0.80; 0.83 and 0.93, respectively) (Table 3).

In the liver, among the eight genes evaluated in the Cobb 500® commercial line, *RPL5*, *EEF1* and *ACTB* were highly expressed (Ct = 18.18; 20.38; 20.75, respectively). The other genes showed moderate expression, with Ct values ranging from 24 to 30.03. However, all genes presented high CV measures and SD values above 1.0, and were, therefore, considered non-stable genes (Table 4).

In the liver of Peloco, the *RPL5* gene showed a high expression (Ct = 16,26), while the other genes had moderate expressions. Regarding CV, *RPL5* and *TFRC* were the genes with least dispersion (CV = 4.86% and 4.3% respectively) compared to the other genes. According to SD measures, only *RPL5* gene showed a deviation lower than 1.0, being therefore considered stable in this tissue (Table 5).

## Expression stability of reference genes

Data were divided into two groups for each tissue (genetic group—Peloco and Cobb 500®—and environment—comfort and acute heat stress), so that stability analyses could cover all factors. The three algorithms (BestKeeper, geNorm and NormFinder) and the RankAggreg tool were used to analyze RG stabilities in both heart (Tables 6–9) and liver (Tables 10–13) tissues.

**Table 3. Descriptive statistics of reference genes expression levels obtained by BestKeeper in the heart tissue of Peloco backyard chicken submitted to comfort (23˚C) and acute heat stress (39.5˚C) conditions.**

| n = 12 | RPL5 | LDHA | MRPS30 | EEF1 | ACTB | HPRT1 | MRPS27 | TFRC |
|---|---|---|---|---|---|---|---|---|
| Geometric mean [Ct] | 17.25 | 26.07 | 27.44 | 21.18 | 19.85 | 24.82 | 25.36 | 29.51 |
| Arithmetic mean [Ct] | 17.27 | 26.09 | 27.47 | 21.20 | 19.89 | 24.85 | 25.39 | 29.54 |
| Min [Ct] | 15.21 | 24.29 | 24.41 | 19.93 | 17.97 | 23.45 | 22.91 | 21.20 |
| Max [Ct] | 18.83 | 27.95 | 29.56 | 22.88 | 22.59 | 27.29 | 27.71 | 31.65 |
| Standard deviation [± Ct] | 0.80 | 1.03 | 1.23 | 0.83 | 1.01 | 0.93 | 1.02 | 1.06 |
| CV [%Ct] | 4.66 | 3.97 | 4.50 | 3.93 | 5.08 | 3.75 | 4.04 | 3.61 |
| Correlation coeff. [r] | 0.66 | 0.82 | 0.77 | 0.80 | 0.70 | 0.77 | 0.83 | 0.32 |

[Ct]: Cycle threshold; Min [Ct] and Max [Ct]: Cycle threshold minimum and maximum values, respectively; CV [%Ct]: Coefficient of variation of Ct levels in percentage; [r]: correlation coefficient.

**Table 4. Descriptive statistics of reference genes expression levels obtained by BestKeeper in the liver tissue of Cobb 500® commercial line submitted to comfort (23˚C) and acute heat stress (39.5˚C) conditions.**

| n = 12 | RPL5 | LDHA | MRPS30 | EEF1 | ACTB | HPRT1 | MRPS27 | TFRC |
|---|---|---|---|---|---|---|---|---|
| **Geometric mean [Ct]** | 18.18 | 24.60 | 25.86 | 20.38 | 20.75 | 29.64 | 27.07 | 30.03 |
| **Arithmetic mean [Ct]** | 19.25 | 24.70 | 25.92 | 20.44 | 20.81 | 29.67 | 27.20 | 30.07 |
| **Min [Ct]** | 17.54 | 21.82 | 23.69 | 18.55 | 17.95 | 28.06 | 24.35 | 27.76 |
| **Max [Ct]** | 22.96 | 28.58 | 29.55 | 24.11 | 23.13 | 32.25 | 34.30 | 32.36 |
| **Standard deviation [± Ct]** | 1.29 | 2.07 | 1.40 | 1.42 | 1.16 | 1.21 | 2.14 | 1.41 |
| **CV [%Ct]** | 6.71 | 8.38 | 5.59 | 6.95 | 5.59 | 4.08 | 7.89 | 4.71 |
| **Correlation coeff. [r]** | 0.84 | 0.85 | 0.75 | 0.89 | 0.88 | 0.47 | 0.60 | 0.40 |

[Ct]: Cycle threshold; Min [Ct] and Max [Ct]: Cycle threshold minimum and maximum values, respectively; CV [%Ct]: Coefficient of variation of Ct levels in percentage; [r]: correlation coefficient.

In the heart tissue, according to the BestKeeper tool, the best RG were *EEF1* (0.73) and *RPL5* (0.80) for Cobb 500® and Peloco, respectively. Considering the environment, the best genes were *RPL5* (0.52) and *LDHA* (0.96) for comfort and stress conditions, respectively. In the BestKeeper ranking considering all factors, *LDHA* gene was indicated as the most stable, whereas *ACTB* was considered the least stable (Table 6).

Using geNorm tool, the RG were classified as follows: for the genetic group, the most stable genes were *RPL5/EEF1* (0.58) and *LDHA/RPL5* (0.66) for Cobb 500® and Peloco, respectively. Regarding the environment, the best genes were *EEF1/RPL5* (0.53) and *EEF1/RPL5* (0.70), for comfort and stress conditions, respectively (Table 7).

Applying the NormFinder algorithm, the most stable genes were *RPL5* (0.46) and *EEF1* (0.58) for Cobb 500® and Peloco, respectively, and *RPL5* (0.30) and *LDHA* (0.57) for comfort and heat stress environments, respectively (Table 8).

The RanKAggreg package [25] ranks the genes from the most stable to the least stable, taking into account the stability values and the frequency at which each gene appears according to the stability analysis tool algorithms (BestKeeper, Genorm and NormFinder). According to the RG overall stability ranking obtained by RankAgregg, the *RPL5* and *ACTB* genes were considered the most and the least stable RG in the heart tissue. (Table 9).

For liver tissue, the genes indicated as most stable using BestKeeper tool were *ACTB* (1.16) and *RPL5* (0.94) for Cobb 500® and Peloco genetic groups, respectively. Regarding the environment, the most stable genes were *RPL5* (0.949) and *TFRC* (0.92), for comfort and acute heat stress, respectively (Table 10).

**Table 5. Descriptive statistics of reference genes expression levels obtained by BestKeeper in the liver tissue of Peloco backyard chicken submitted to comfort (23˚C) and acute heat stress (39.5˚C) conditions.**

| n = 12 | RPL5 | LDHA | MRPS30 | EEF1 | ACTB | HPRT1 | MRPS27 | TFRC |
|---|---|---|---|---|---|---|---|---|
| **Geometric mean [Ct]** | 16.26 | 26.06 | 27.42 | 21.46 | 22.7 | 30.30 | 28.64 | 30.40 |
| **Arithmetic mean [Ct]** | 19.29 | 26.12 | 27.49 | 21.54 | 22.79 | 30.36 | 28.71 | 30.44 |
| **Min [Ct]** | 17.55 | 22.48 | 24.29 | 18.81 | 19.30 | 28.14 | 25.23 | 27.95 |
| **Max [Ct]** | 22.28 | 29.69 | 32.17 | 25.36 | 26.50 | 34.63 | 32.89 | 32.75 |
| **Standard deviation [± Ct]** | 0.93 | 1.46 | 1.46 | 1.46 | 1.71 | 1.50 | 1.56 | 1.30 |
| **CV [%Ct]** | 4.86 | 5.62 | 5.32 | 6.77 | 7.51 | 4.95 | 5.44 | 4.30 |
| **Correlation coeff. [r]** | 0.94 | 0.92 | 0.95 | 0.89 | 0.87 | 0.90 | 0.96 | 0.70 |

[Ct]: Cycle threshold; Min [Ct] and Max [Ct]: Cycle threshold minimum and maximum values, respectively; CV [%Ct]: Coefficient of variation of Ct levels in percentage; [r]: correlation coefficient.

**Table 6. BestKeeper rankings with reference genes stability values considering each factor (genetic group and environment) and all factors in chicken heart tissue.** Values in parentheses indicate the genes ranking by factor.

| Genes | Cobb n = 12 | Peloco n = 12 | Comfort n = 12 | Stress n = 12 | BestKeeper Ranking |
|-------|-------------|---------------|----------------|---------------|---------------------|
| *ACTB* | 1.14 (5) | 1.01 (4) | 1.13 (8) | 1.70 (8) | 8 |
| *EEF1* | **0.73 (1)** | 0.83 (2) | 0.67 (2) | 1.01 (3) | 4 |
| *HPRT1* | 1.67 (8) | 0.93 (3) | 1.10 (7) | 1.31 (7) | 5 |
| *LDHA* | 1.04 (4) | 1.03 (6) | 0.80 (3) | **0.96 (1)** | 1 |
| *MRPS27* | 1.16 (6) | 1.02 (5) | 0.86 (4) | 1.24 (6) | 3 |
| *MRPS30* | 0.75 (3) | 1.23 (8) | 0.88 (5) | 1.07 (4) | 6 |
| *RPL5* | 0.73 (2) | **0.80 (1)** | **0.52 (1)** | 1.01 (2) | 2 |
| *TFRC* | 1.16 (7) | 1.06 (7) | 1.06 (6) | 1.14 (5) | 7 |

Using geNorm tool, the most stable genes were *EEF1/RPL5* (0.76) and *EEF1/LDHA* (0.63) for Cobb 500® and Peloco, respectively. Regarding the environment, the most stable genes were *EEF1/LDHA* (0.80 and 0.82) for both comfort and acute heat stress conditions, respectively (Table 11).

Applying the NormFinder algorithm, the genes indicated as most stable in Cobb 500® and Peloco genetic groups were *ACTB* (0.48) and *RPL5* (0.65), respectively. Considering the environment, the most stable genes were *ACTB* (0.54) and *MRPS30* (0.74) for comfort and acute heat stress, respectively (Table 12).

According to the RankAgregg package, the most and least stable RG were *RPL5* and *MRPS27*, respectively, in liver tissue (Table 13).

## Discussion

The use of suitable reference genes to normalize RT-qPCR data is essential for obtaining results that actually represent the relative abundance of gene transcripts in different species, cells and tissues [24].

In this study, *RPL5*, *EEF1* and *LDHA* genes were ranked as the most stable in the heart tissue of the studied genetic groups. Contrasting the overall ranking, in the other conditions (Cobb, Peloco, comfort and heat stress), the most stable genes varied among *RPL5*, *EEF1* and *LDHA* and the least stable was the *ACTB* gene. This result corroborates the findings of Cedraz et al. [21], in which the authors reported that *EEF1* and *RPL5* genes were classified as the most stable in chickens exposed to high temperatures. Nascimento et al. [19] also reported that *ACTB* gene was the least stable in *Gallus gallus* Pectoralis muscle. The low *ACTB* gene stability in the current study suggests that its regulation may be affected by the experimental

**Table 7. GeNorm rankings with reference genes stability values considering each factor (genetic group and environment) and all factors in chicken heart tissue.** Values in parentheses indicate the genes ranking by factor.

| Genes | Cobb n = 12 | Peloco n = 12 | Comfort n = 12 | Stress n = 12 | geNorm Ranking |
|-------|-------------|---------------|----------------|---------------|-----------------|
| *ACTB* | 1.15 (6) | 1.16 (7) | 1.22 (8) | 1.53 (8) | 8 |
| *EEF1* | **0.58 (1)** | 0.78 (3) | **0.53 (1)** | **0.70 (1)** | 2 |
| *HPRT1* | 1.40 (8) | 1.03 (5) | 0.90 (5) | 1.24 (6) | 7 |
| *LDHA* | 0.85 (4) | **0.66 (1)** | 0.54 (3) | 0.85 (4) | 4 |
| *MRPS27* | 1.29 (7) | 1.11 (6) | 1.11 (7) | 1.15 (5) | 5 |
| *MRPS30* | 0.74 (3) | 0.81 (4) | 0.67 (4) | 0.81 (3) | 3 |
| *RPL5* | **0.58 (1)** | **0.66 (1)** | **0.53 (1)** | **0.70 (1)** | 1 |
| *TFRC* | 0.97 (5) | 1.23 (8) | 1.03 (6) | 1.36 (7) | 6 |

**Table 8. NormFinder rankings with reference genes stability values considering each factor (genetic group and environment) and all factors in chicken heart tissue.** Values in parentheses indicate the genes ranking by factor.

| Genes | Cobb n = 12 | Peloco n = 12 | Comfort n = 12 | Stress n = 12 | NormFinder Ranking |
|-------|-------------|---------------|----------------|---------------|--------------------|
| *ACTB* | 1.44 (6) | 0.98 (6) | 1.48 (8) | 1.91 (8) | 6 |
| *EEF1* | 0.48 (2) | **0.58 (1)** | 0.34 (2) | 0.64 (2) | 1 |
| *HPRT1* | 1.61 (8) | 0.80 (3) | 1.21 (6) | 1.32 (6) | 8 |
| *LDHA* | 0.58 (3) | 0.76 (2) | 0.63 (3) | **0.57 (1)** | 3 |
| *MRPS27* | 1.24 (5) | 0.84 (4) | 0.82 (4) | 1.18 (5) | 4 |
| *MRPS30* | 0.69 (4) | 1.12 (7) | 0.89 (5) | 0.93 (4) | 5 |
| *RPL5* | **0.46 (1)** | 0.88 (5) | **0.30 (1)** | 0.89 (3) | 2 |
| *TFRC* | 1.51 (7) | 1.46 (8) | 1.39 (7) | 1.60 (7) | 7 |

conditions, since it was considered a suitable normalizer in other studies [35,19]. Such expression variations can be observed within the same tissue performing its physiological functions [26], since the amount of transcripts may differ in comfort and heat stress situations, as well as in different genetic groups.

In Peloco heart tissue, the *RPL5* gene was classified as the most stable in two of the three algorithms applied, and it was the gene with the highest expression. One of the desirable features of a suitable normalizer reference gene is its high expression, i.e., low Ct value, since the Ct value is dependent on the amount of molecules presented at the beginning of the amplification process [27]. This result corroborates the findings of Cedraz et al. [21], in which the *RPL5* remained among the most stable genes in Peloco pectoral muscle in both comfort and acute heat stress conditions. On the other hand, in the heart tissue of Cobb 500® commercial line, the *EEF1* gene presented the highest stability in two of the three analyzed tools. These variations are expected to occur, since there are differences between tissues of divergent genetic groups, as well as among different statistical algorithms [21].

The *RPL5* gene encodes a small protein, component of ribosomal 60S subunit and responsible for transporting 5S rRNA to the nucleus. This protein acts specifically with the casein kinase II beta subunit and is typical for genes encoding ribosomal proteins [28]. In this sense, *RPL5* is in frequent action, since it is involved in the essential process of cell rRNA transport, which partly explains its greater expression in the studied chicken tissues. Mengmeng et al. [29] reported that *RPL5* was the most stable gene in human heart tissue, whereas *ACTB* was the least stable.

The *EEF1* gene encodes a protein responsible for the alpha1 elongation factor and is involved in the enzymatic delivery of aminoacyl tRNAs to the ribosome during protein synthesis. Thus, its expression is considered continuous, which may explain its stability in the heart

**Table 9. Overall ranking of reference genes in heart tissue obtained with the different tools (Bestkeeper, geNorm and NormFinder) and ranked by the RankAggreg package.**

| | BestKeeper Ranking | GeNorm Ranking | NormFinder Ranking | RankAggreg Overall ranking |
|-------|--------------------|-----------------|--------------------|-----------------------------|
| *ACTB* | 8 | 8 | 6 | 8 |
| *EEF1* | 4 | 2 | 1 | 2 |
| *HRPT1* | 5 | 7 | 8 | 6 |
| *LDHA* | 1 | 4 | 3 | 3 |
| *MRPS27* | 3 | 5 | 4 | 5 |
| *MRPS30* | 6 | 3 | 5 | 4 |
| *RPL5* | 2 | 1 | 2 | 1 |
| *TFRC* | 7 | 6 | 7 | 7 |

**Table 10. BestKeeper rankings with reference genes stability values considering each factor (genetic group and environment) and all factors in chicken liver tissue.**
Values in parentheses indicate the genes ranking by factor.

| Genes | Cobb n = 12 | Peloco n = 12 | Comfort n = 12 | Stress n = 12 | BestKeeper Ranking |
|---|---|---|---|---|---|
| *ACTB* | **1.16 (1)** | 1.71 (8) | 1.39 (3) | 1.84 (8) | 5 |
| *EEF1* | 1.42 (5) | 1.5 (3) | 1.52 (5) | 1.41 (4) | 3 |
| *HPRT1* | 1.21 (2) | 1.51 (6) | 1.47 (4) | 1.09 (2) | 4 |
| *LDHA* | 2.07 (7) | 1.47 (5) | 1.97 (7) | 1.67 (6) | 7 |
| *MRPS27* | 2.15 (8) | 1.56 (7) | 2.47 (8) | 1.71 (7) | 8 |
| *MRPS30* | 1.45 (6) | 1.46 (4) | 1.92 (6) | 1.53 (5) | 6 |
| *RPL5* | 1.29 (3) | **0.94 (1)** | **0.95 (1)** | 1.25 (3) | 2 |
| *TFRC* | 1.42 (4) | 1.31 (2) | 1.26 (2) | **0.92 (1)** | 1 |

tissue of chickens subjected to heat stress. In the study of Kishore et al. [30], the *EEF1* gene was the seventh most stable in buffalo under heat stress conditions among 11 RG analyzed.

The *LDHA* gene participates in glycolysis process. The protein encoded by this gene catalyzes the conversion of L-lactate and NAD to pyruvate and NADH in the final step of anaerobic glycolysis [28]. This protein is predominantly found in muscle tissue, and in birds, it is especially active in erythrocytes. As *LDHA* is part of an important chemical reaction that provides energy to the organism, its expression is constant, which may explain its stability in heart tissue [31, 32].

In the liver, the most stable genes in the overall ranking, i.e., including all factors (Cobb 500® and Peloco; comfort and acute heat stress), were *RPL5*, *EEF1* and *ACTB*. The *RPL5* and *ACTB* genes were considered the most stable in two of the three algorithms in Peloco and Cobb 500® liver, respectively. In this tissue, *MRPS27* was indicated as the least stable reference gene. This result differs from the findings of Cedraz et al. [21], in which the authors reported that *MRPS27* was the most stable gene in breast muscular tissue of Peloco genetic group.

*ACTB* is the gene that encodes one of the six existing actin proteins, involved in cell motility, structure and integrity [28]. In birds, its expression is greater in heart, kidney, liver, brain and skeletal muscle [33]. Although expressed in all tissues, great variability for this gene was found in some studies with birds, limiting its use as a reference gene, despite its high expression [34,19].

Several RT-qPCR studies have sought to validate RG in different species, tissues and treatments in animals, including cattle [14], chickens [18, 19, 20,21], pigs [15], sheep [35], horses [36], birds [18, 19, 5, 37], and fish [16,17], as well as in plants [38,39]. It is noteworthy that there is not a universal reference gene, and analyzing several factors in different tissues and organisms that are constantly adapting to changing conditions, different RG expression

**Table 11. GeNorm rankings with reference genes stability values considering each factor (genetic group and environment) and all factors in chicken liver tissue.**
Values in parentheses indicate the genes ranking by factor.

| Genes | Cobb n = 12 | Peloco n = 12 | Comfort n = 12 | Stress n = 12 | GeNorm Ranking |
|---|---|---|---|---|---|
| *ACTB* | 1.29 (5) | 1.10 (7) | 1.19 (5) | 1.38 (8) | 7 |
| *EEF1* | **0.76 (1)** | **0.63 (1)** | **0.80 (1)** | **0.82 (1)** | 1 |
| *HPRT1* | 1.51 (6) | 1.03 (6) | 1.34 (6) | 1.26 (6) | 6 |
| *LDHA* | 0.93 (3) | **0.63 (1)** | **0.80 (1)** | **0.82 (1)** | 2 |
| *MRPS27* | 1.85 (8) | 0.89 (4) | 1.70 (8) | 0.99 (3) | 5 |
| *MRPS30* | 1.09 (4) | 0.98 (5) | 0.92 (3) | 1.19 (5) | 4 |
| *RPL5* | **0.76 (1)** | 0.81 (3) | 1.04 (4) | 1.09 (4) | 3 |
| *TFRC* | 1.62 (7) | 1.18 (8) | 1.44 (7) | 1.33 (7) | 8 |

**Table 12. NormFinder rankings with reference genes stability values considering each factor (genetic group and environment) and all factors in chicken liver tissue.** Values in parentheses indicate the genes ranking by factor.

| Genes | Cobb n = 12 | Peloco n = 12 | Comfort n = 12 | Stress n = 12 | NormFinder Ranking |
|-------|-------------|---------------|----------------|---------------|--------------------|
| ACTB | **0.48 (1)** | 1.15 (7) | **0.54 (1)** | 1.31 (7) | 3 |
| EEF1 | 0.84 (2) | 0.94 (6) | 0.73 (3) | 0.93 (3) | 1 |
| HPRT1 | 1.48 (5) | 0.80 (5) | 1.22 (4) | 0.93 (3) | 7 |
| LDHA | 1.61 (6) | 0.76 (4) | 1.41 (6) | 1.11 (5) | 5 |
| MRPS27 | 2.56 (8) | 0.68 (2) | 2.55 (8) | 0.77 (2) | 8 |
| MRPS30 | 1.37 (4) | 0.69 (3) | 1.41 (6) | **0.74 (1)** | 4 |
| RPL5 | 0.99 (3) | **0.65 (1)** | 0.67 (2) | 1.17 (6) | 2 |
| TFRC | 1.70 (7) | 1.35 (8) | 1.39 (5) | 1.36 (8) | 6 |

profiles can be observed [40]. Therefore, it is extremely important to consider unique approaches and RG validation for each experiment separately. Moreover, considering the gene expression stability rankings obtained by different algorithms, it should always be a priority finding a high stable gene considering all possible algorithms [27].

In addition, it is important to highlight that there are genes with tissue-specific expressions, which are under specific regulation, as reported by Bentz et al. [41], who found differential genes expression among tissues in swallows. For example, muscle-specific genes were associated with muscle contraction, and spleen-specific genes with immune response. On the other hand, there are different genes controlled by temporal regulation, i.e., they are expressed in certain periods, as reported by Laine et al. [42], in which songbird genes were differentially expressed at different times as well as at different temperature treatments. Therefore, different tissues, as well as different genetic groups subjected to adverse environmental conditions, may express different structural and protein components, resulting in a specific gene profile to suit each necessity [43].

To date, there are no studies evaluating RG in Peloco and Cobb 500® heart and liver tissues under the same experimental conditions used in this study (comfort and acute heat stress). In this way, the results of the current study may be useful for future research with Peloco backyard chicken and Cobb 500® commercial line, since studies performed so far mainly focused on phenotypic traits. In addition, it is important to analyze the expression profile of RG in specific tissues, as they may influence the interpretation of the analyzed data.

Therefore, correct normalization is indispensable, since it is crucial to take into account the biological relevance of different species and/or tissues samples, as well as the experimental conditions. In this sense, the three most stable reference genes found in this study for heart and liver tissues of chickens subjected to comfort and acute heat stress conditions are adequate to

**Table 13. Overall ranking of reference genes in liver tissue obtained with the different tools (Bestkeeper, geNorm and NormFinder) and ranked by the RankAggreg package.**

| | BestKeeper Ranking | GeNorm Ranking | NormFinder Ranking | RankAggreg Overall ranking |
|-------|--------------------|----------------|--------------------|----------------------------|
| ACTB | 5 | 7 | 3 | 3 |
| EEF1 | 3 | 1 | 1 | 2 |
| HRPT1 | 4 | 6 | 7 | 6 |
| LDHA | 7 | 2 | 5 | 4 |
| MRPS27 | 8 | 5 | 8 | 8 |
| MRPS30 | 6 | 4 | 4 | 5 |
| RPL5 | 2 | 3 | 2 | 1 |
| TFRC | 1 | 8 | 6 | 7 |

normalize gene expression data from different chicken genetic groups that exhibit divergent behavior when induced by heat stress.

## Conclusion

According to the RankAggreg tool classification based on the three different algorithms (Best-Keeper, geNorm and NormFinder), the most stable reference genes were *RPL5*, *EEF1* and *LDHA* for heart tissue and *RPL5*, *EEF1* and *ACTB* for liver tissue of chickens subjected to comfort and acute heat stress environmental conditions. However, the best reference genes may vary depending on the experimental conditions of each study, such as different breeds, environmental stressors, and tissues analyzed. Therefore, the need to perform priori studies to assay the best reference genes at the outset of each study is emphasized.

## Supporting information

**S1 Data.**
(XLS)

## Author Contributions

**Conceptualization:** Juliana Gracielle Gonzaga Gromboni, Haniel Cedraz de Oliveira, Antônio Amândio Pinto Garcia Junior, Ronaldo Vasconcelos Farias Filho, Caio Fernando Gromboni, Amauri Arias Wenceslau.

**Data curation:** Juliana Gracielle Gonzaga Gromboni, Haniel Cedraz de Oliveira, Antônio Amândio Pinto Garcia Junior.

**Formal analysis:** Juliana Gracielle Gonzaga Gromboni, Haniel Cedraz de Oliveira, Daniele Botelho Diniz Marques, Antônio Amândio Pinto Garcia Junior.

**Funding acquisition:** Juliana Gracielle Gonzaga Gromboni, Haniel Cedraz de Oliveira, Antônio Amândio Pinto Garcia Junior, Ronaldo Vasconcelos Farias Filho, Amauri Arias Wenceslau.

**Investigation:** Juliana Gracielle Gonzaga Gromboni, Haniel Cedraz de Oliveira, Antônio Amândio Pinto Garcia Junior, Teillor Machado Souza, Amauri Arias Wenceslau.

**Methodology:** Juliana Gracielle Gonzaga Gromboni, Haniel Cedraz de Oliveira, Antônio Amândio Pinto Garcia Junior, Caio Fernando Gromboni, Teillor Machado Souza, Amauri Arias Wenceslau.

**Project administration:** Juliana Gracielle Gonzaga Gromboni, Antônio Amândio Pinto Garcia Junior, Ronaldo Vasconcelos Farias Filho, Amauri Arias Wenceslau.

**Resources:** Juliana Gracielle Gonzaga Gromboni, Antônio Amândio Pinto Garcia Junior, Ronaldo Vasconcelos Farias Filho, Amauri Arias Wenceslau.

**Software:** Juliana Gracielle Gonzaga Gromboni, Haniel Cedraz de Oliveira, Caio Fernando Gromboni.

**Supervision:** Juliana Gracielle Gonzaga Gromboni, Antônio Amândio Pinto Garcia Junior, Ronaldo Vasconcelos Farias Filho, Caio Fernando Gromboni, Amauri Arias Wenceslau.

**Validation:** Juliana Gracielle Gonzaga Gromboni, Haniel Cedraz de Oliveira, Daniele Botelho Diniz Marques, Antônio Amândio Pinto Garcia Junior, Caio Fernando Gromboni.

**Visualization:** Juliana Gracielle Gonzaga Gromboni, Daniele Botelho Diniz Marques, Caio Fernando Gromboni.

**Writing – original draft:** Juliana Gracielle Gonzaga Gromboni, Daniele Botelho Diniz Marques, Caio Fernando Gromboni, Amauri Arias Wenceslau.

**Writing – review & editing:** Juliana Gracielle Gonzaga Gromboni, Daniele Botelho Diniz Marques, Caio Fernando Gromboni.

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
