## [Decision Letter · Decision Letter 0]

5 Nov 2019

PONE-D-19-27029

Influence of heat stress on reference genes stability in heart and liver of commercial and backyard chickens

PLOS ONE

Dear Dr Gromboni,

Thank you for submitting your manuscript to PLOS ONE. After careful consideration, we feel that it has merit but does not fully meet PLOS ONE’s publication criteria as it currently stands. Therefore, we invite you to submit a revised version of the manuscript that addresses the points raised during the review process.

We would appreciate receiving your revised manuscript by Dec 20 2019 11:59PM. To enhance the reproducibility of your results, we recommend that if applicable you deposit your laboratory protocols in protocols.io, where a protocol can be assigned its own identifier (DOI) such that it can be cited independently in the future. For instructions see: http://journals.plos.org/plosone/s/submission-guidelines#loc-laboratory-protocols

We look forward to receiving your revised manuscript.

Kind regards,

Michael H. Kogut, Ph.D.

Academic Editor

PLOS ONE

Journal Requirements:

2. We note that you are reporting an analysis of a microarray, next-generation sequencing, or deep sequencing data set. PLOS requires that authors comply with field-specific standards for preparation, recording, and deposition of data in repositories appropriate to their field. Please upload these data to a stable, public repository (such as ArrayExpress, Gene Expression Omnibus (GEO), DNA Data Bank of Japan (DDBJ), NCBI GenBank, NCBI Sequence Read Archive, or EMBL Nucleotide Sequence Database (ENA)). In your revised cover letter, please provide the relevant accession numbers that may be used to access these data. For a full list of recommended repositories, see http://journals.plos.org/plosone/s/data-availability#loc-omics or http://journals.plos.org/plosone/s/data-availability#loc-sequencing.

"This study was financially supported by Fundação de Amparo à Pesquisa no Estado da Bahia (FAPESB) - 0348/2015, Universidade Estadual de Santa Cruz (UESC) and Universidade Estadual do Sudoeste da Bahia (UESB).".

i) Please provide an amended statement that declares *all* the funding or sources of support (whether external or internal to your organization) received during this study, as detailed online in our guide for authors at http://journals.plos.org/plosone/s/submit-now.  Please also include the statement “There was no additional external funding received for this study.” in your updated Funding Statement.

ii) Please include your amended Funding Statement within your cover letter. We will change the online submission form on your behalf.

Reviewers' comments:

Reviewer's Responses to Questions

**Comments to the Author**

1. Is the manuscript technically sound, and do the data support the conclusions?

Reviewer #1: Yes

Reviewer #2: Yes

2. Has the statistical analysis been performed appropriately and rigorously? 

Reviewer #1: No

Reviewer #2: Yes

3. Have the authors made all data underlying the findings in their manuscript fully available?

Reviewer #1: Yes

Reviewer #2: No

4. Is the manuscript presented in an intelligible fashion and written in standard English?

Reviewer #1: Yes

Reviewer #2: Yes

5. Review Comments to the Author

Reviewer #1: Title could be modified to read "Influence of heat stress on reference genes stability in heat and liver of two chicken genotypes

Line79-80: Delete

Line 84 insert "quantitative" between Real-time and polymerase.

Line 86 Replace sentence with "As RT-qPCR is"...

Line 126 Replace "birth" with "hatch"

Line 131 Delete "(wood chips)"

Line 137 should be modified to read "with ad libitum access to water and feed". Delete In a second moment.

Please ages of the chickens when they were euthanized.

Line 139-141. Please rephrase sentence. Observation cannot prevent death.

How was heart rate measured?

Line 142 Change "slaughtered" to "euthanized"

Line 147 Change "slaughter" to "euthanization" and "fragments" to "tissues"

Line 167 Delete "in literature"

Line 174/200 Change real-time..PCR to RT-qPCR

Line 184 Change EAU to USA

Analysis:

The results show a clear genotype by treatment interaction and the results have to be presented as such: Cobb (comfort), Cobb (Heat stress), Peloco (comfort) and Peloco (Heat stress). The authors subsequently make a case for that. Line412-414 "the amount of transcripts may differ in comfort and heat stress, as well as in different genetic groups"

RESULTS:

Line 253-255 Delete or move to discussion. You are expected to present results from the current study. Also, Line 265-266.

Line 264. Insert Table 3 after respectively.

It is not clear how Tables 2-5 were generated. They show expression in a tissue of a genotype in two environments. Are the values in the two environments averaged? or one is expressed over the other? Authors should explain what the values represent with respect to the two temperatures.

Discussion:

Line 406-408 The results in the current study is not in agreement as different tissues were used. The sentence could be restated as "Nascimento et al. [19] also reported that ACTB gene was the least stable in Gallus gallus Pectoralis muscle.

Line 433 RPL5 second to?

Line 441 EEF1 second most stable compared to?

Line 462 Replace "impairing" to "limiting"

Line 474. Provide examples and references

Comfort can be changed to Thermoneutral throughout the text.

Reviewer #2: The manuscript by Gromboni et al reports a study to identy reference genes in two tissues for assaying heat stress in chicken. While the study is conducted very well, and reported very clearly, the effort required to identify reference genes - clearly an exercise required for every study - basically argues that this type of approach is required for every study with any variation in breeds, stressors, or tissue sources. This conclusion then greatly limits the utility of this report, except arguing for the need for a priori studies to assay RG at the outset of each study. In this reviewers opinion, these should be at least, among the main conclusions of this report.

L134: This reviewer has a bit of hesitation accepting that 39.5c is the only metric applied for heat stressed. The temperature is quite a bit off from the higher limits of the chicken thermoneutral zone. While it is reasonable to assume that elevated temperatures cause thermal stress, as far as a phenotypic classification for assessing stability of transcripts, this critiera is insufficient. They mention behavior as a stress marker, but if they had either a serum corticosterone or HSP titer, this categorization would be appropriate. Without this type of secondary confirmation of a heat stress state, we are just looking at expression stability at two differen ambient temperature ranges. Similarly for the ‘comfort’ state. If we are to accept these a phenotypic states, they need to be clearly defined.

L154: What was the RNA quality? What criteria were applied to determine if quality was sufficient for RT-qPCR?

L167: Why were these reference genes selected? Please add additional details.

L184: Please change ‘EUA’ to ‘USA’

Table 1: Please provide either the ENSEMBL gene id or transcript id. The IDs reported in this table are not correct ensembl ids.

L202: Non template controls were not used. This is an important experimental design oversight.

With so many markers used, please provide additional details on the plate set-up of the reactions. Were comfort time samples included on all plates?

L497: While the methodology and the analysis show that the RPL5 , EEF1, and LDHA genes are the best reference genes for assessing heat stress in these tissues, the main takeaway for this reviewer hava been that a) reference genes may have to be evaluated for every study based on breed, stressor, and tissue sampled, and b) the use of the RankAggreg tool is required to come to this decision. These efforts suggest that the amount of work required to identify a correct RG is extremely high, in proportion to the value of the information revealed. Overall, to this reviewer, this paper argues against the use of qPCR methods for assaying phenotypic states in chicken tissues.

L506: misspelled.

6. PLOS authors have the option to publish the peer review history of their article (what does this mean?). If published, this will include your full peer review and any attached files.

Reviewer #1: No

Reviewer #2: No

---

## [Author Response · Author response to Decision Letter 0]

12 Jan 2020

We thank the editorial board for your interest in our manuscript entitled “INFLUENCE OF HEAT STRESS ON REFERENCE GENES STABILITY IN HEART AND LIVER OF COMMERCIAL AND BACKYARD CHICKENS”. We are pleased to have the opportunity to return the revised manuscript to Plos One for appreciation: "INFLUENCE OF HEAT STRESS ON REFERENCE GENES STABILITY IN HEART AND LIVER OF TWO CHICKENS GENOTYPES". Together with the revised version of the manuscript, we include below a point-by-point reply to the comments of reviewers and details of the modifications made.

---

## [Editor Report · Decision Letter 1]

14 Jan 2020

Influence of heat stress on reference genes stability in heart and liver of two chickens genotypes.

PONE-D-19-27029R1

Dear Dr. Gromboni,

We are pleased to inform you that your manuscript has been judged scientifically suitable for publication and will be formally accepted for publication once it complies with all outstanding technical requirements.

With kind regards,

Michael H. Kogut, Ph.D.

Academic Editor

PLOS ONE
---

## [Editor Report · Acceptance letter]

23 Jan 2020

PONE-D-19-27029R1 

Influence of heat stress on reference genes stability in heart and liver of two chickens genotypes. 

Dear Dr. Gromboni:

I am pleased to inform you that your manuscript has been deemed suitable for publication in PLOS ONE. Congratulations! Your manuscript is now with our production department. 

With kind regards,

on behalf of

Dr. Michael H. Kogut 

Academic Editor

PLOS ONE